# Rotational Projection Errors in Coronal Knee Alignment on Weight-Bearing Whole-Leg Radiographs: A 3D CT Reference Across CPAK Morphotypes

**DOI:** 10.3390/bioengineering12080794

**Published:** 2025-07-23

**Authors:** Igor Strahovnik, Andrej Strahovnik, Samo Karel Fokter

**Affiliations:** 1Faculty of Medicine, University of Ljubljana, Kongresni trg 12, 1000 Ljubljana, Slovenia; igor.strahovnik@gmail.com (I.S.); andrej_strahovnik@yahoo.com (A.S.); 2Valdoltra Orthopeadic Hospital, Jadranska cesta 31, 6280 Ankaran, Slovenia; 3University Medical Center Maribor, Ljubljanska ulica 5, 2000 Maribor, Slovenia; 4Faculty of Medicine, University of Maribor, Slomškov trg 15, 2000 Maribor, Slovenia

**Keywords:** CPAK classification, MDFA, coronal alignment, weight-bearing radiographs, 3D CT, projection-based alignment index, systematic projection error, femoral notch projection ratio (FNPR) index, total knee arthroplasty

## Abstract

Whole-leg radiographs (WLRs) are widely used to assess coronal alignment before total knee arthroplasty (TKA), but may be inaccurate in patients with atypical morphotypes or malrotation. This study evaluated the discrepancy between WLR and 3D computed tomography (CT) scans across coronal plane alignment of the knee (CPAK) morphotypes and introduced a novel projection index—the femoral notch projection ratio (FNPR). In CPAK III knees, 19% of cases exceeded a clinically relevant threshold (>3° difference), prompting investigation of underlying projection factors. In 187 knees, coronal angles—including the medial distal femoral angle (MDFA°), medial proximal tibial angle (MPTA°), femoral mechanical angle (FMA°), and arithmetic hip–knee–ankle angle (aHKA°)—were measured using WLR and CT. Rotational positioning on WLR was assessed using FNPR and the patellar projection ratio (PPR). CPAK classification was applied. WLR systematically underestimated alignment, with the greatest bias in CPAK III (MDFA° + 1.5° ± 2.0°, *p* < 0.001). FNPR was significantly higher in CPAK III and VI (+1.9° vs. −0.3°, *p* < 0.001), indicating a tendency toward internally rotated limb positioning during imaging. The PPR–FNPR mismatch peaked in CPAK III (4.1°, *p* < 0.001), suggesting patellar-based centering may mask rotational malprojection. Projection artifacts from anterior osteophytes contributed to outlier measurements but were correctable. Valgus morphotypes with oblique joint lines (CPAK III) were especially prone to projection error. FNPR more accurately reflected rotational malposition than PPR in morphotypes prone to patellar subluxation. A 3D method (e.g., CT) or repeated imaging may be considered in CPAK III to improve surgical planning.

## 1. Introduction

Total knee arthroplasty (TKA) is currently the most widely used and cost-effective treatment for end-stage knee osteoarthritis [1]. Despite its widespread success, up to 20% of patients report suboptimal outcomes, frequently attributed to residual malalignment or altered joint biomechanics [2,3]. Conventional mechanical alignment, aimed at restoring neutral coronal alignment, may not be optimal for all patients—especially those with constitutional varus or valgus morphology. Studies show that up to one-third of healthy individuals have non-neutral alignment, and this percentage increases with age and degenerative changes [4,5,6]. Optimal coronal alignment in TKA remains a subject of ongoing debate, with traditional approaches including mechanical, anatomical, and kinematic alignment each offering distinct benefits and limitations.

In response to the variability in native knee phenotypes, a novel classification system known as the coronal plane alignment of the knee (CPAK) was introduced by MacDessi et al. [7] to better characterize coronal knee morphology. The CPAK system integrates two key anatomical parameters—the arithmetic mechanical hip–knee–ankle angle (aHKA) and the joint line obliquity (JLO)—to define nine distinct knee types. This stratification allows for a more individualized assessment of coronal knee anatomy and provides a framework for tailoring alignment strategies in TKA.

In all these approaches, accurate preoperative imaging remains essential [8,9,10,11,12]. While weight-bearing whole-limb radiographs (WLRs) are the current standard for evaluating coronal alignment, their reliability can be compromised by rotational malposition, joint deformities, or patient-specific anatomy. Other imaging options such as computed tomography (CT), magnetic resonance imaging (MRI), low-dose biplanar imaging (EOS), or navigation are less standardized or accessible.

A major limitation of 2D radiographic imaging is its susceptibility to projection bias resulting from rotational malposition during acquisition—an often overlooked but frequent source of error [13,14]. Such bias can significantly distort coronal angle measurements, particularly in valgus morphotypes with oblique joint lines, where standard patellar-based centering becomes unreliable due to patellar subluxation or femoral torsion.

To address this, we introduce the femoral notch projection ratio (FNPR), a novel radiographic index that estimates rotational projection bias directly from femoral condylar geometry. The FNPR is derived from the radiograph itself and is independent of patellar tracking, offering a practical intra-modality indicator of rotational image alignment.

This study evaluates the accuracy of 2D WLR-based coronal angle assessment compared to 3D CT across CPAK morphotypes, and explores the potential of FNPR to identify systematic projection bias caused by rotational malposition during image acquisition.

## 2. Materials and Methods

### 2.1. Study Design and Cohort

This study is a sub-analysis of a larger prospective cohort evaluating coronal alignment accuracy in patients undergoing total knee arthroplasty (TKA). The full cohort and imaging protocol have been registered on ClinicalTrials.gov (NCT05277883) and were approved by the national ethics board (ref. no. 0120-252/2021/6).

The present analysis focuses on the impact of CPAK morphotypes on radiographic accuracy and projection error, representing an independent hypothesis derived from the same imaging and morphometric dataset.

All participants provided written informed consent. Exclusion criteria were prior lower limb surgery, inflammatory joint disease, or fixed flexion deformity >15°.

### 2.2. Imaging Protocol

Radiographs: WLR were acquired in a single-leg stance using a digital system (Ysio, Siemens Healthcare, Erlangen, Germany). The patella was centered over the femoral condyles; source-to-detector distance was 300 cm. Images from the hip, knee, and ankle were digitally stitched into a composite whole-leg image [15,16,17].

CT Scans: Bilateral lower-limb CT scans were performed using a 140 kVp protocol (IQon Spectral, Philips Healthcare, Best, The Netherlands), with a slice thickness of 1.5 mm and an approximate dose of 1.5 mSv [18]. Patients were scanned supine, with extended and neutrally rotated limbs. CT images were reconstructed using a standard bone algorithm kernel. All images were processed in 3D Slicer (v5.3.0; Kitware, Inc., Clifton Park, NY, USA) [19] for landmark-based measurements.

### 2.3. Landmark Definition and Measurement

Two observers performed all measurements; discrepancies >1.5° (coronal) or >3° (axial) were resolved by consensus. Observers were blinded to clinical data and radiographic classifications during measurement to reduce bias.

The term projection error describes the angular misrepresentation of anatomical axes on 2D radiographs, resulting from rotational malposition, joint line obliquity, and the lack of depth perception inherent to planar imaging. CT was used as the geometric reference standard for all comparisons, given its reproducibility and freedom from projection bias.

Anatomical landmarks were defined per established protocols [20,21,22] and identified semi-automatically in 3D Slicer (Figure 1). All coronal angles were derived exclusively from bony landmarks and were selected for their low sensitivity to postural or soft-tissue variation.

Angular parameters assessed in the coronal plane included the mechanical medial distal femoral angle (MDFA°), the mechanical medial proximal tibial angle (MPTA°), and the angle between the anatomical and mechanical femoral axes (FMA°). The arithmetic hip–knee–ankle angle (aHKA°) was calculated by summing the MDFA° and MPTA° [23]. Patients were stratified into nine morphotypes using a modified version of the CPAK classification [7], which used MDFA° instead of lateral distal femoral angle (LDFA°) for defining joint line obliquity (Figure 2). For the purposes of this study, CPAK types III and VI—both characterized by valgus alignment—were collectively referred to as valgus morphotypes.

The center of the femoral head was calculated by fitting a sphere to manually selected surface points. The femoral neck and diaphyseal axes were defined by best-fit lines through their respective centers. Femoral joint surface points (sF line in Figure 1) were defined as the most distal point of the central articular surface of the medial and lateral femoral condyles.

To minimize parallax errors on radiographs, lateral and medial central tibial plateau point was defined at the midpoint between anterior and posterior margins, improving consistency of MPTA° measurements (Figure 3) [24]. For CT images, the centers of the medial and lateral tibial plateaus for definition of sT line were determined using three manually selected points on each plateau.

The ankle joint center was defined as the midpoint of the line connecting the tips of the medial and lateral malleoli.

To estimate lower limb rotational positioning on WLR, two projection-based indices were employed: the patellar projection ratio (PPR) and the femoral notch projection ratio (FNPR). PPR, adapted from Moon et al. [12], quantifies the deviation of the patellar center from the midpoint of the femoral trans-epicondylar line. Because this measurement may be affected by patellar subluxation, we introduced FNPR, defined as the normalized horizontal deviation of the femoral notch apex (Nx) from the midpoint of the femoral trans-epicondylar line. The calculation was performed using the following formula:FNPR = ((Nx − ((E1x + E2x)/2))/|E1x − E2x|) × 100
where E1x and E2x represent the horizontal (x-axis) coordinates of the medial and lateral femoral epicondyles, respectively. The FNPR = 0 position corresponds to neutral rotational projection, where the femoral notch apex lies equidistant between the medial and lateral epicondyles. This approximates the coronal plane of the distal femoral flexion-extension axis, often considered the ideal reference for neutral rotational alignment. Positive FNPR values indicate internal rotational projection, while negative values reflect external rotational projection relative to this axis (Figure 4). This symmetric reference was used as a practical and anatomically justified threshold for identifying malprojection.

To estimate the clinical meaning of FNPR values, we simulated anteroposterior (AP) projections of the distal femur on five 3D CT cases under controlled axial rotations (±20°, in 5° increments). FNPR values were calculated for each projection and compared to the known degree of limb rotation to establish a functional calibration.

Both PPR and FNPR were normalized to the length of the trans-epicondylar line, enabling direct comparison. As projection-dependent indices, both FNPR and PPR reflect apparent limb rotation and do not represent actual anatomical torsion.

The PPR–FNPR mismatch served as an indirect indicator of rotational malposition on WLR. Larger mismatches suggest a greater disconnect between patellar tracking and true femoral orientation, consistent with projection bias—especially in knees with patellar subluxation.

Since limb positioning commonly relies on patellar centering, such mismatches can introduce unintended rotational malposition, potentially compromising the accuracy of coronal alignment measurements.

Lower leg flexion was assessed only on CT, by measuring the sagittal components of mechanical axes vectors of the femur and tibia. Additionally, femoral torsion (tF°) was defined as the angle between the femoral neck axis and the posterior condylar line. Tibial torsion (tT°) was defined as the angle between the posterior tibial condylar line and the transmalleolar axis. Angles were designated as positive when the distal segment showed internal rotation. Femoral bowing was calculated as the deviation between the mid-diaphyseal axis and the geometric center of the femoral shaft at the same level.

### 2.4. Statistical Analysis

All statistical analyses were performed using IBM SPSS Statistics for Windows, version 22.0 (IBM Corp., Armonk, NY, USA).

Group comparisons were performed using the Wilcoxon signed-rank test for paired non-parametric data. Kruskal–Wallis tests were used for comparisons across CPAK subgroups. Outlier analysis was performed using a predefined clinical threshold of 3°, in accordance with Paley’s recommendation [25].

Multiple linear regression models were used to assess the association between coronal alignment CT and WLR measurement projection discrepancy (e.g., CPAK type, PPR, FNPR, femoral bowing). Sensitivity analysis confirmed consistency with unilateral data. While the present analysis focused on conventional inferential statistics, future modelling could benefit from adaptive regularization techniques for variable selection in complex, high-dimensional datasets [26,27].

## 3. Results

### 3.1. Demographic and Interrater Analysis

A total of 187 knees from 112 patients were included in the final analysis. Patients with a history of prior contralateral surgery were excluded, including 12 with previous TKA, 11 with total hip arthroplasty, one with hip osteosynthesis following a fracture, and one with a tibial tuberosity transfer. In total, 25 patients were excluded based on these criteria.

Patient demographics is shown in Table 1 and CPAK types distribution in Figure 5.

A high degree of inter- and intra-observer reliability was observed for all angular measurements, consistent with findings reported in previous studies [28] (Table 2). The highest agreement was observed for FMA° on CT (ICC = 0.95). MDFA° on WLR had the lowest ICC, consistent with its higher sensitivity to projection variation.

### 3.2. Coronal Alignment Parameters Analysis

Using measured epicondylar distances and notch displacements, we established that one unit of FNPR corresponds to approximately 3–5° of axial rotational projection on CT-based simulations. However, this calibration cannot be directly applied to WLR-derived FNPR values due to projection geometry differences. In particular, the parallax effect inherent to WLR may exaggerate FNPR values. Therefore, this CT-based calibration should be regarded as an upper-bound estimate, providing a crude interpretative range for FNPR values as surrogates for internal or external rotational projection during radiographic acquisition. Values above 5 may indicate >15–20° of internal rotational projection, potentially compromising coronal angle measurements on WLR.

On WLR, the average PPR was −1.5, indicating a slight external projection of the limb. In contrast, the average FNPR was +0.16, suggesting a mild internal rotational malposition during image acquisition.

When stratified by CPAK classification, valgus morphotypes III and VI showed higher FNPR values (+1.9 and +0.6, respectively; combined average +1.4), in contrast to all other CPAK types, which demonstrated negative FNPR values (combined average −0.6). This difference was statistically significant (*t*-test, *p* < 0.001).

The largest external PPR values were observed in CPAK types III, V, and VI (−2.2, −2.2, and −3.8, respectively), while CPAK IV showed a slight internal rotation (PPR = +0.8).

CPAK types III and VI exhibited the highest PPR–FNPR mismatch, with a mean absolute difference of 4.2 versus 0.0 in all other morphotypes (*p* < 0.001). This pronounced divergence between patellar- and notch-based indicators reflects a valgus-specific projection pattern, suggesting increased vulnerability to rotational malprojection during WLR acquisition.

Statistically significant measurement discrepancies were observed between CT and WLR measurements for all coronal alignment parameters (all *p* < 0.05), as summarized in Table 3. Although this difference was systematic, it remained within 1° on average across all parameters, with WLR consistently underestimating the values.

The largest mean measurement discrepancy was observed for FMA° (1.0° ± 0.8°, *p* < 0.001). In contrast, MPTA° demonstrated minimal difference and low variability (0.2° ± 1.0°), indicating excellent agreement and few outliers.

Both aHKA° and MDFA° exhibited moderate discrepancies (1.0° ± 1.8° and 0.8° ± 1.7°, respectively), along with increased variability. Given that aHKA° is calculated from MDFA° and MPTA°, and the latter remained consistent, the observed offset in aHKA° likely stems from projection-related deviation in MDFA°.

### 3.3. Correlation Between FNPR and CT-Based Torsional Parameters

We evaluated whether FNPR reflects underlying anatomical rotation by correlating with 3D CT-derived torsional parameters. Results are summarized in Table 4.

### 3.4. Measurement Analysis by CPAK Groups

Stratification of MDFA° values between CT and WLR by CPAK morphotype revealed statistically significant variation across groups. In contrast, no significant differences were observed for aHKA°, MPTA°, or FMA° among CPAK types.

The MDFA° discrepancy was most pronounced in CPAK type III, with a mean difference of +1.5° ± 2.0°, whereas CPAK V demonstrated near-zero divergence (−0.0° ± 1.1°; *p* = 0.005 for post hoc comparison). Detailed results are provided in Table 5.

When CPAK groups were clustered by joint line obliquity (JLO), morphotypes with distal apex obliquity (CPAK I–III) demonstrated significantly greater MDFA° measurement difference (+1.1° ± 1.8°) compared to those with neutral JLO (CPAK IV–VI; +0.3° ± 1.4°, *p* < 0.001), as illustrated in Figure 6.

### 3.5. Outlier Analysis

Bland–Altman outlier analysis demonstrated good agreement between CT and WLR measurements for MPTA° and FMA°, with low rates of statistical outliers (5.3% and 4.8%, respectively) and clinical outliers (1.1% and 1.6%, respectively).

In contrast, aHKA° and MDFA° showed moderate clinical outlier rates, with 12.8% and 10.2% of cases exceeding the predefined ±3° threshold, despite acceptable statistical outlier levels (5.9% and 6.4%).

When CPAK morphotypes were grouped by joint line obliquity, distal apex types (CPAK I–III) showed a significantly higher incidence of MDFA° deviations >3° (14.5%) compared to neutral apex types (CPAK IV–VI) (2.9%) (Fisher’s exact test, *p* = 0.012; OR = 5.61). Among individual morphotypes, CPAK III exhibited the highest rate of clinically significant deviation (19%), nearly three times greater than the combined rate in all other types (7.5%; OR = 2.87). These findings are summarized in Table 6.

To further explore this pattern, targeted comparisons between CPAK III and all other morphotypes revealed that CPAK III had significantly higher FNPR values (+1.8° vs. –0.3°, *p* < 0.001) and a more pronounced PPR–FNPR mismatch (−4.1° vs. −1.0°, *p* < 0.001), consistent with rotational malposition during radiographic acquisition. MDFA° values measured on CT were also significantly greater in CPAK III (95.7° vs. 91.9°, *p* < 0.001), indicating a more valgus-oriented distal femur. No significant differences were noted in PPR alone or in knee flexion angle.

A subgroup analysis of MDFA° clinical outliers (>3°) identified distinct anatomical and positional contributors. These cases showed significantly lower knee flexion during imaging (2.2° vs. 6.2°, *p* < 0.001) and substantially elevated FNPR values (+2.1° vs. −0.1°, *p* = 0.009), suggesting internal malprojection. Although mean PPR and MDFA° values were also higher in outliers, they did not reach statistical significance (*p* = 0.059 for both).

Similar trends were observed in the aHKA° analysis, which reflects combined femoral and tibial geometry. Distal apex types (CPAK I–III) had a 3.3-fold increased likelihood of exceeding the ±3° threshold (*p* = 0.039), with CPAK III again showing the highest rate of projection-related deviation (28.6% vs. 7.5% in other types; *p* = 0.001; OR = 4.43), underscoring its susceptibility to projection distortion.

Visual inspection of the 19 MDFA° clinical outliers identified a distinct “double contour” artifact in 15 cases, caused by anterior femoral osteophytes projecting into the joint space due to internal rotation. In these cases, the inferior contour was often misidentified as the lateral condylar edge, resulting in overestimation of the MDFA° in the varus direction.

Upon remeasurement using a standardized approach that consistently selected the superior contour (Figure 7), the average projection error in these cases decreased from 4.8° to 2.4°. After correction, only 6 cases remained above the 3° threshold, with mean residual deviation reduced to 3.5° and maximum deviation lowered from 8.0° to 4.2°.

### 3.6. Subgroup Analysis by Sex and Age

Subgroup analyses were conducted to assess the impact of sex and age on leg positioning and projection error during WLR imaging (Table 7).

Significant sex-based differences were observed in both projection indices and MDFA° measurement accuracy. Female patients exhibited greater internal femoral rotation (FNPR mean = +0.5 vs. −0.4) and higher MDFA° bias (1.1 vs. 0.4), with a clinical outlier rate nearly four times that of male patients (12.7% vs. 3.3%). Interestingly, the mismatch between patellar and femoral notch orientation (PPR–FNPR) was more pronounced in males (−2.0 vs. −1.4), indicating potential sex-related differences in rotational control during limb positioning.

Similarly, age was found to be an influential factor. Younger patients (<67 years) had the highest mean MDFA° bias (1.1°) and outlier rate (12.2%), while patients over 72 years exhibited the lowest projection error (0.7°) and lowest outlier rate (1.7%).

Collectively, these findings suggest that both sex and age influence rotational positioning and the reliability of coronal alignment measurements on standing WLR.

Female and younger individuals appeared more susceptible to inconsistent limb positioning, potentially reflecting greater joint laxity or variable soft-tissue dynamics, emphasizing the need for enhanced standardization protocols in these subgroups.

## 4. Discussion

### 4.1. Accuracy of WLR and Morphotype-Specific Projection Errors

Our study confirmed that while WLR are generally reliable for assessing coronal alignment, notable projection error may occur in specific morphotypes, particularly those with valgus alignment and distal joint line obliquity (CPAK III). Among the coronal parameters, FMA° exhibited the largest systematic discrepancy between CT and WLR, though with low variance and few outliers. In contrast, MDFA° and aHKA° showed higher variability and clinically relevant outliers exceeding 3°, particularly in CPAK types I–III.

In CPAK III, 19% of knees showed an MDFA° difference > 3° between CT and WLR. A dedicated subgroup analysis of cases with large MDFA° CT and WLR discrepancies (>3°) revealed distinct anatomical and biomechanical characteristics. These outliers demonstrated significantly lower knee flexion at the time of CT imaging, indicating greater extension and suggesting increased joint laxity. They also exhibited significantly higher FNPR values and FNPR–PPR mismatch, consistent with internal rotational malposition during WLR acquisition, likely related to patellar subluxation.

Visual inspection revealed that anterior osteophytes can project onto the joint line, particularly if limb is internally malpositioned (higher FNPR), creating a double contour that leads to underestimation of the true articular surface. Adjusting for this artefact reduced the outlier rate by more than half.

### 4.2. Comparison with Prior Literature

Previous studies have shown good inter-observer reliability for both WLR and CT-based methods in assessing coronal alignment [10,15,28,29,30,31]. Our findings align with recent comparisons of WLR, CT, and EOS, which reported only modest differences across modalities, often attributable to weight-bearing effects and rotational malposition [14,29,32,33,34]. Leon Muñoz et al. suggested that discrepancies in hip–knee–ankle angle (HKA°) reflect soft-tissue instability, while differences in MDFA° and MPTA° indicate rotation during WLR acquisition [15,35]. We extend this analysis by quantifying rotational discrepancy using FNPR and PPR, demonstrating that internal rotational projection errors are most pronounced in valgus morphotypes, likely exacerbated by patellar subluxation.

Research indicates that coronal alignment is associated with distinct phenotypic differences in the lower limb [36,37]. Valgus alignment has been linked to increased distal internal femoral rotation and lateral condyle hypoplasia [38,39], while varus alignment is often correlated with increased femorotibial external rotation [40,41,42]. These variations likely contribute to the projection discrepancies observed.

### 4.3. FNPR as an Indicator of Rotational Malprojection

Rotational positioning on WLR is influenced not only by acquisition technique but also by intrinsic knee biomechanics. In valgus knees, an increased Q-angle exerts greater lateral force on the patella, predisposing to lateral patellar subluxation during quadriceps contraction [43]. This lateral displacement shifts the patellar landmark, and when radiographs are centered on the patella—as commonly done in clinical practice—the limb may be inadvertently positioned in internal rotation, resulting in projection-based misalignment.

The femoral notch projection ratio (FNPR) provides a notch-based estimate of rotational projection, independent of patellar tracking. In our cohort, FNPR values were predominantly positive in CPAK types III and VI, where valgus alignment and patellar subluxation are more prevalent. In CPAK III specifically, the average FNPR was +1.87, with 71% of cases showing positive values and a mean of +3.26 among those with FNPR > 0. By comparison, other CPAK types had lower proportions of positive FNPR (e.g., 54% in CPAK II, 39% in CPAK I), and their mean values among positive cases were also lower (1.5–2.1).

These findings suggest that both the frequency and magnitude of internal rotational projection are greater in valgus morphotypes—particularly CPAK III. In this group, 19% of cases were MDFA outliers, and 88% of those showed positive FNPR values, supporting the role of internal malprojection as a contributing factor. These internal rotational tendencies were also associated with greater MDFA° measurement discrepancy on WLR compared to CT, consistent with malprojection due to patellar-based centering.

As further evidence of this mechanism, the mismatch between patellar-based (PPR) and notch-based (FNPR) indices was most pronounced in valgus morphotypes, particularly CPAK III, which exhibited the largest average PPR–FNPR mismatch (−4.1). This divergence suggests that the patella and the distal femur are not aligned in these cases, supporting the interpretation that patellar-based centering leads to rotationally biased imaging.

### 4.4. Future Directions

Beyond imaging accuracy, future research should investigate how rotational projection patterns—particularly internal malprojection in valgus morphotypes—impact functional outcomes after total knee arthroplasty (TKA). Stratifying patients based on coronal morphotype (e.g., CPAK III) and radiographic indicators of internal projection (e.g., positive FNPR) may help identify subgroups at higher risk for suboptimal biomechanics, implant malposition, or long-term failure.

Specifically, studies could explore whether positive FNPR values in CPAK III knees are associated with unintended varus resections in distal femur cuts, femoral component malrotation on postoperative CT scans, joint line obliquity mismatch, or lower patient-reported outcomes including pain, instability, or dissatisfaction.

FNPR may serve as a preoperative quality control indicator, highlighting possible rotational malposition when markedly positive. For example, valgus knees with evident internal projection on WLR may benefit from repeated imaging with adjusted positioning or, when warranted, rotationally validated modalities such as CT to ensure accurate alignment measurement and optimize surgical planning.

In parallel, machine learning-based annotation pipelines could enable automated, reproducible assessment of FNPR and coronal alignment parameters from WLR, supporting routine use in clinical workflows. Such tools may not only improve diagnostic precision but also pave the way for more personalized surgical planning, accounting for individual rotational and morphological variations. Ultimately, integrating AI-assisted quality control measures into the preoperative workflow may enhance alignment accuracy and contribute to improved long-term outcomes.

### 4.5. Limitations

This study compares weighted WLR with non-weight-bearing CT, potentially introducing bias. Although CT provides high anatomical accuracy, it may underestimate joint laxity or dynamic varus/valgus deviation under axial load. Previous studies have reported angular differences of up to 1.5–2.0° between these modalities, particularly in knees with increased ligamentous laxity or valgus alignment [15,27,33]. We focused exclusively on geometric angles derived from bone landmarks, which are less influenced by axial compression or soft-tissue laxity. This mitigates some concerns about posture-induced discrepancies. Nonetheless, we acknowledge that functional deviations under load may introduce alignment differences not captured by CT. Anatomical landmarking was semi-automatic and operator-dependent. FNPR is novel and requires external validation. The cohort included only patients with advanced osteoarthritis (Kellgren–Lawrence III–IV), limiting generalizability. No postoperative or functional outcomes were included. Flexion was measured only on CT, with no standardized assessment available for WLR, which may introduce unmeasured variability. Finally, although bilateral clustering was addressed via sensitivity analysis, residual clustering may persist.

## 5. Conclusions

This study confirmed that WLR systematically underestimates coronal alignment compared to 3D CT, with the greatest MDFA° discrepancies observed in valgus morphotypes—especially CPAK III. These deviations were linked to internal rotational malprojection during imaging, likely due to patellar subluxation and patella-based centering. FNPR served as a useful indicator of projection bias. Recognizing such malprojection—particularly in younger, female, or valgus-aligned patients—may improve radiographic interpretation and guide the need for repeat imaging or rotationally validated modalities during preoperative planning.

### Clinical Recommendation

We recommend incorporating coronal morphotype classification (e.g., CPAK) into radiographic protocols to personalize alignment assessment and improve diagnostic accuracy in total knee arthroplasty. FNPR may be used descriptively to assess rotational image quality, particularly in valgus knees. When internal malprojection is suspected—especially in CPAK III or in younger and female patients—repeat imaging or the use of validated rotational imaging such as CT may help avoid interpretation bias and support accurate surgical planning.

## Figures and Tables

**Figure 1 bioengineering-12-00794-f001:**
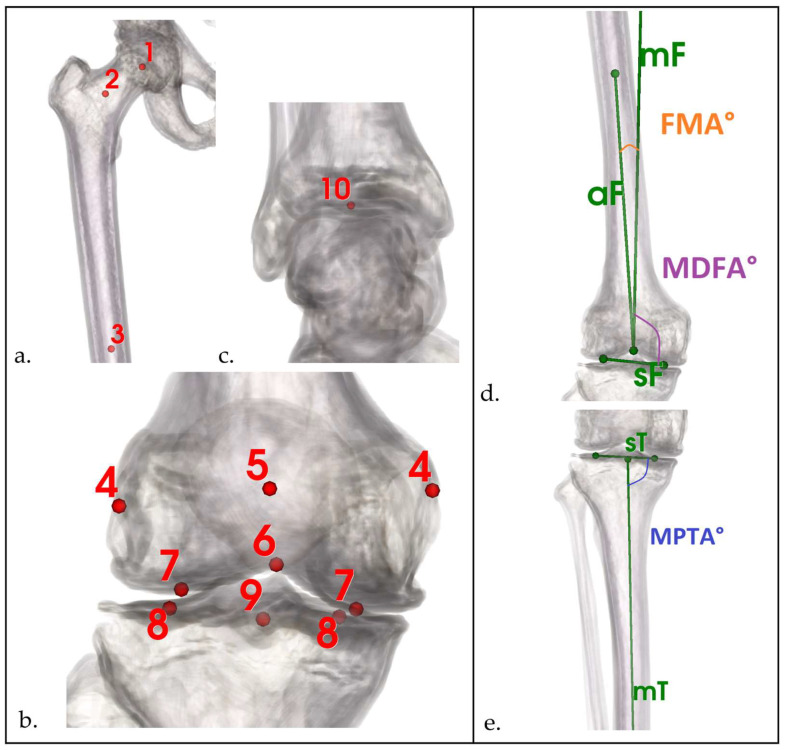
Definition of key anatomical points and axis determination used for 2D and 3D angle measurements. (**a**–**c**) Key anatomical landmarks: (1) femoral head center, (2) femoral neck, (3) femoral shaft, (4) epicondylar axis, (5) patellar center, (6) femoral notch, (7,8) distal femoral and tibial plateau centers, (9) knee center, (10) ankle center. (**d**,**e**) Derived axes and angles: anatomical and mechanical axes of femur (aF, mF), distal femoral and proximal tibial surfaces (sF, sT), femoral mechanical angle (FMA°), medial distal femoral angle (MDFA°), medial proximal tibial angle (MPTA°).

**Figure 2 bioengineering-12-00794-f002:**
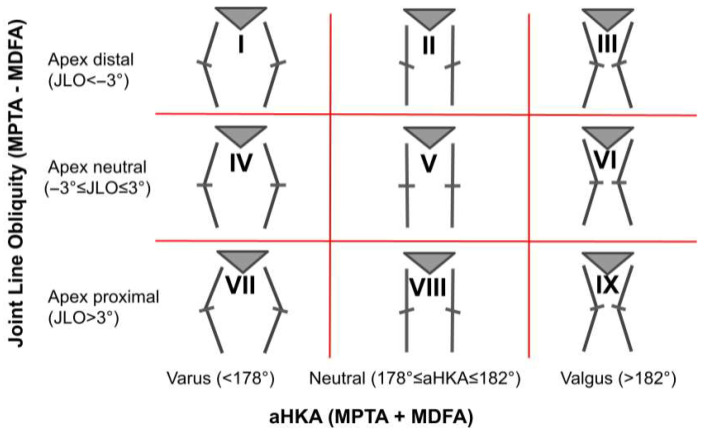
CPAK classification is based on joint line obliquity (JLO) and arithmetical hip–knee–ankle angle (aHKA°), with 9 morphotypes. Modified for use with mechanical medial distal femoral angle (MDFA°) instead of mechanical lateral distal femoral angle (LDFA°). MPTA° = mechanical proximal tibial angle.

**Figure 3 bioengineering-12-00794-f003:**
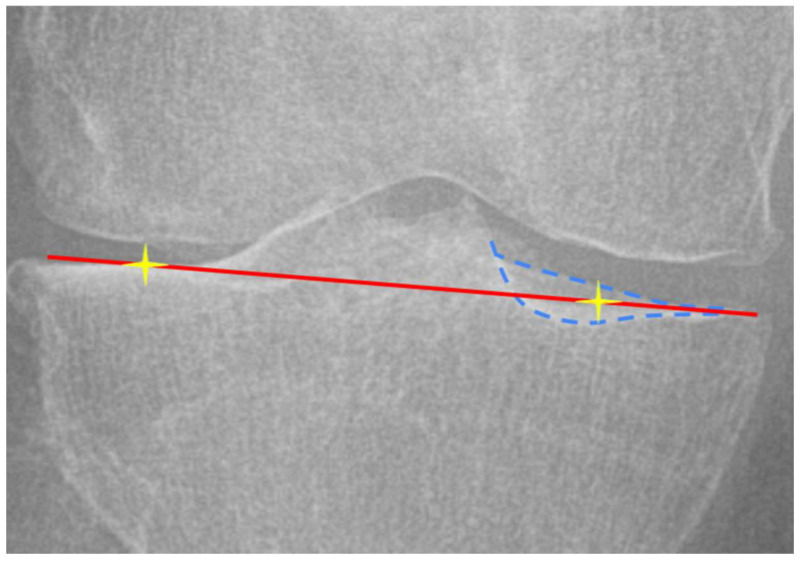
Parallax artifact in mechanical medial proximal tibial angle (MPTA°) measurement on weight-bearing whole-limb radiographs (WLRs). Blue dashed lines: anterior and posterior plateau margins; Red line: MPTA° angle; Yellow star: central tibial plateau point used for consistent angulation definition.

**Figure 4 bioengineering-12-00794-f004:**
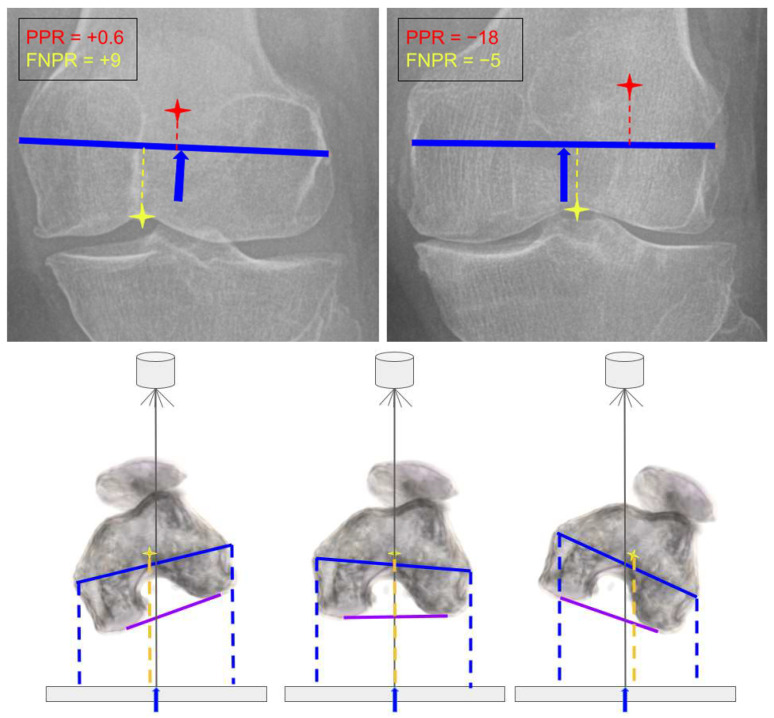
Patellar projection ratio (PPR) and femoral notch projection ratio (FNPR). (**Upper images**): PPR (red) indicates the projected position of the patella; FNPR (yellow) marks the apex of the femoral notch; the blue line represents the midline reference. **Left panel**: Positive values reflect medial deviation of the projection point, corresponding to internal (inward) limb rotation. **Right panel**: Negative values indicate lateral deviation, corresponding to external (outward) limb rotation. (**Lower images**)**:** Axial representation of the femoral notch apex in simulated internal rotation (**left**), external rotation (**right**), and neutral projection (**center**), with the apex (yellow star) centered on the epicondylar line (blue line). Notch deviation on WLR appears exaggerated compared to CT simulations due to parallax effects.

**Figure 5 bioengineering-12-00794-f005:**
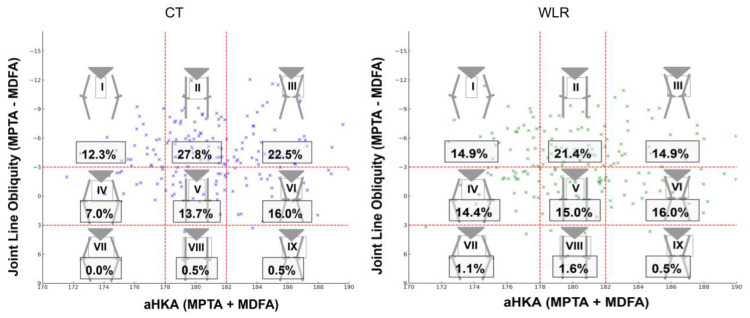
Distribution of Coronal Plane Alignment of the Knee (CPAK) groups for computer tomography (CT) and weight-bearing whole-limb radiographs (WLRs). aHKA° = arithmetical hip–knee–ankle angle; MPTA° = mechanical proximal tibial angle. MDFA° = mechanical medial distal femoral angle. Blue and green markers indicate individual data points for CT and WLR measurements, respectively. Red dashed lines define the CPAK classification boundaries.

**Figure 6 bioengineering-12-00794-f006:**
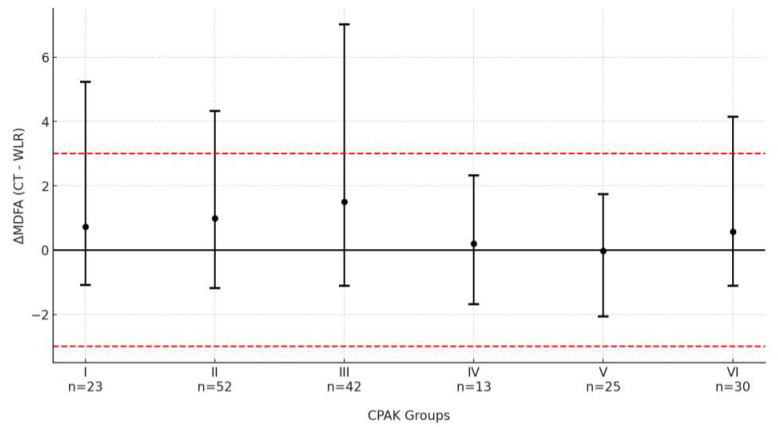
MDFA° measurement difference (CT − WLR) across CPAK morphotypes. Dots represent group means; brackets indicate 95% confidence intervals; red dashed line shows ±3° clinical outlier threshold.

**Figure 7 bioengineering-12-00794-f007:**
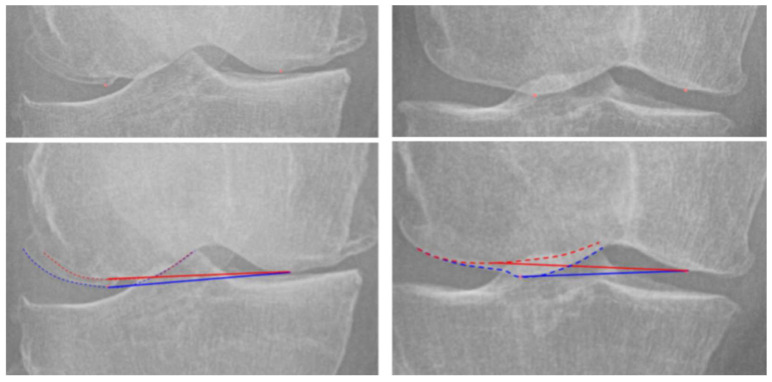
Example of double contour artifact in whole-limb radiograph (WLR) image. (**Upper picture**): without annotations; (**lower picture**): with annotation; Blue dashed line: osteophyte projection; Red dashed line: true articular surface; Solid line: selected measurement contour. Using upper contour reduces projection error.

**Table 1 bioengineering-12-00794-t001:** Patient demographics and CPAK classification based on CT and WLR imaging.

Patient Demographics		
Parameter	Value	
Number of knees	187	
Age, years (mean [95% CI])	69.7 [68.4–70.8]	
Age group (*n*)		
- <67	46	
- 67–72	62	
- >72	79	
Gender, *n* (F/M)	107/80	
Side, *n* (right/left)	93/94	
Kellgren–Lawrence grade (2/3/4)	20/55/112	
**CPAK Classification:**		
**CPAK Type**	**CT (*n*)**	**WLR (*n*)**
I	23	28
II	52	40
III	42	28
IV	13	27
V	25	28
VI	30	30
VII	0	2
VIII	1	3
IX	1	1

CI = confidence interval; F/M = female/male; CPAK = coronal plane alignment of the knee; CT = computed tomography; WLR = weight-bearing whole-limb radiograph.

**Table 2 bioengineering-12-00794-t002:** Inter- and intra-observer intraclass correlation coefficients (ICC) for CT and WLR-based angle measurements.

	Inter-Observer		Intra-Observer
Parameter	ICC (CT)	ICC (WLR)	ICC (WLR)
aHKA°	0.81	0.78	0.80
MDFA°	0.80	0.76	0.76
MPTA°	0.88	0.85	0.86
FMA°	0.95	0.90	0.94
FNPR	-	0.70	0.76

Intraclass correlation coefficients (ICC) reflect interobserver agreement for measurements performed on CT and weight-bearing whole-limb radiographs (WLRs). Intra-observer agreement was assessed on WLR for 30 cases by one reader. All ICCs were calculated using a two-way random-effects model for absolute agreement. ICC > 0.75 indicates good reliability; ICC > 0.90 indicates excellent reliability. aHKA° = arithmetical hip–knee–ankle angle; MDFA° = mechanical distal femoral angle; MPTA° = mechanical proximal tibial angle; FMA° = angle between the femoral mechanical and anatomical axes; FNPR = femoral notch projection ratio.

**Table 3 bioengineering-12-00794-t003:** Comparison of coronal alignment angles measured on CT and WLR using the Wilcoxon signed-rank test.

Angle	CT	WLR	∆ (CT − WLR)	*p*-Value
aHKA°	181.3° ± 3.1°	180.4° ± 3.2°	1° ± 1.8°	<0.001
MDFA°	92.7° ± 1.8°	91.9° ± 1.9°	0.8° ± 1.7°	<0.001
MPTA°	88.6° ± 1.6°	88.5° ± 1.6°	0.2° ± 1.0°	0.028
FMA°	6.6° ± 1.0°	5.6° ± 1.0°	1.0° ± 0.8°	<0.001

Values are presented as mean ± standard deviation. *p*-values were calculated using the Wilcoxon signed-rank test. ∆ = measurement difference (CT − WLR), expressed in degrees. aHKA° = arithmetical hip–knee–ankle angle; MDFA° = mechanical medial distal femoral angle; MPTA° = mechanical medial proximal tibial angle; FMA° = angle between the femoral mechanical and anatomical axes.

**Table 4 bioengineering-12-00794-t004:** Pearson correlations between FNPR and CT torsion metrics.

Torsional Parameter	Pearson r	*p*-Value
Femoral torsion (tF)	0.150	0.040
Tibial torsion (tT)	−0.209	0.001
Knee torsion	−0.237	0.001
Total limb torsion	±0.204	0.005

**Table 5 bioengineering-12-00794-t005:** Distribution of MDFA° measurement difference (CT − WLR) by CPAK morphotype, assessed with the Wilcoxon signed-rank test.

CPAK Type	*n*	∆MDFA°	SD	*p*-Value
I	23	+0.7°	1.7°	0.092
II	52	+1.0°	1.5°	<0.001
III	42	+1.5°	2.0°	<0.001
IV	13	+0.2°	1.4°	0.735
V	25	–0.0°	1.1°	0.958
VI	30	+0.6°	1.6°	0.177

∆MDFA° = measurement difference (CT − WLR). Positive values indicate higher MDFA° on CT. *p*-values were obtained from one-sample Wilcoxon signed-rank tests comparing each group’s mean difference to zero.

**Table 6 bioengineering-12-00794-t006:** MDFA° outlier rates (statistical and clinical) across CPAK morphotypes.

CPAK Group	*n*	Outliers (±1.96 SD)	Outliers (±3°)
I	23	2 (8.7%)	2 (8.7%)
II	52	2 (3.8%)	7 (13.5%)
III	42	5 (11.9%)	8 (19.0%)
IV	13	0 (0.0%)	0 (0.0%)
V	25	1 (4.0%)	0 (0.0%)
VI	30	1 (3.3%)	2 (6.7%)
VII	0	–	–
VIII	1	0 (0.0%)	0 (0.0%)
IX	1	0 (0.0%)	0 (0.0%)
All groups	187	11 (5.9%)	19 (10.2%)

Outliers were defined as either exceeding ±1.96 standard deviations from the mean difference (statistical outliers, per the Bland-Altman method) or exceeding ±3° measurement difference (clinical threshold). Only CPAK types I–VI were included in statistical analyses due to low sample size in types VII–IX. MDFA° = mechanical medial distal femoral angle.

**Table 7 bioengineering-12-00794-t007:** Subgroup comparison of FNPR, PPR, MDFA° bias, and outlier rates by sex and age.

Group	FNPR	PPR	∆MDFA°	PPR–FNPR	Outlier > 3°
Female	+0.5	−0.9	1.1	−1.3	12.7%
Male	−0.4	−2.4	0.4	−2.0	3.3%
Age < 67	+0.9	−0.6	1.1	−1.5	12.2%
Age 67–72	−0.3	−1.3	0.6	−1.0	10.3%
Age > 72	−0.4	−3.2	0.7	−2.8	1.7%

FNPR = Femoral Notch Projection Ratio; PPR = Patellar Projection Ratio; ∆MDFA° = Difference between CT and WLR measurements (positive = higher on CT); PPR–FNPR = Mismatch between patellar and notch-based projection indices; Outliers = Percentage of cases exceeding ±3° in MDFA°. Group sizes (*n*) are listed in Table 1.

## Data Availability

The data used in our manuscript is available on request (privacy reason).

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
