# Peer review of "Rotational Projection Errors in Coronal Knee Alignment on Weight-Bearing Whole-Leg Radiographs: A 3D CT Reference Across CPAK Morphotypes"

_bioengineering, 2025, doi:10.3390/bioengineering12080794_

Round 1

Reviewer 1 Report

Comments and Suggestions for Authors
  1. The study suggests clinical implications (e.g., the use of FNPR and CT imaging in CPAK III cases), but it lacks outcome-based validation. Discuss or propose how FNPR-defined malrotation could be linked to surgical accuracy or patient outcomes in future research.
  2. CT scans are non-weight-bearing, whereas WLRs are weight-bearing, which may affect alignment measurements. Acknowledge and discuss this discrepancy, citing prior studies that quantify its potential impact.
  3. FNPR is proposed as a new metric, but no validation or benchmarking against established standards is provided. Justify the use of FNPR and explain the rationale for choosing the 0° threshold.
  4. The current results (e.g., FNPR +1.87) lack clinical relevance without corresponding angular rotation values. Calibrate or relate FNPR/PPR mismatch to approximate degrees of limb rotation to enhance clinical interpretability.
  5. The study relies solely on coronal plane analysis despite having access to 3D CT imaging. Consider analyzing 3D femoral or tibial rotation and correlating it with FNPR values to validate the projection error.
  6. The manuscript contains several grammatical inconsistencies and awkward sentence constructions that affect readability. A thorough language revision is recommended to improve clarity and professional tone.

Author Response

  1. The study suggests clinical implications (e.g., the use of FNPR and CT imaging in CPAK III cases), but it lacks outcome-based validation. Discuss or propose how FNPR-defined malrotation could be linked to surgical accuracy or patient outcomes in future research.
    1. Author Response:
      Thank you for this important observation. We agree that linking radiographic projection quality to surgical outcomes is a critical next step. While this study focuses on imaging accuracy, our findings suggest that internal rotational malprojection—particularly in CPAK III knees—may affect preoperative alignment assessment and component planning.

      In our revised discussion (Section 4.4), we propose that FNPR may serve as a descriptive marker of rotational bias, rather than a strict stratification tool. Future research could evaluate whether positive FNPR values in valgus morphotypes (particularly CPAK 3) correlate with unintended varus resections, femoral component malrotation (as measured on postoperative CT), or joint line obliquity mismatch. Furthermore, prospective studies could assess whether high-FNPR knees exhibit worse patient-reported outcomes, such as pain, instability, or dissatisfaction following total knee arthroplasty.

      We have now clarified this perspective in the Future Directions section and emphasized that FNPR-defined malprojection should be further validated in relation to surgical accuracy and clinical outcomes.

  2. CT scans are non-weight-bearing, whereas WLRs are weight-bearing, which may affect alignment measurements. Acknowledge and discuss this discrepancy, citing prior studies that quantify its potential impact.
    1. Author Response:
      Thank you for this important point. We have now added a paragraph in the Limitations section (4.5) acknowledging the inherent difference between non-weight-bearing CT and weight-bearing WLR. While CT provides a rotationally accurate anatomical reference, the lack of axial load may influence joint space configuration and apparent coronal alignment. Several studies have quantified this discrepancy:

      • Specogna et al. (2007) and Leon-Muñoz et al. (2020) reported that mechanical alignment may differ by up to 1.5°–2° between loaded and unloaded imaging,
      • These differences are more pronounced in knees with joint laxity or valgus alignment.

      Although we attempted to minimize this bias by focusing on geometric parameters less affected by load (e.g., MDFA°, MPTA°), we agree that some variation due to weight-bearing status remains possible and have explicitly acknowledged this in the revised text.

  3. FNPR is proposed as a new metric, but no validation or benchmarking against established standards is provided. Justify the use of FNPR and explain the rationale for choosing the 0° threshold.
    1. Author Response:

      Thank you for this valuable observation. We acknowledge that FNPR is a projection-based index rather than a validated diagnostic parameter. While FNPR = 0 reflects an idealized neutral projection—where the femoral notch lies centered between the epicondyles—we agree that this reference point should not be interpreted as a formal threshold.

      In our revised analysis, we no longer propose FNPR as a strict cutoff tool. Instead, we use FNPR descriptively, to characterize the direction of rotational projection (positive = internal, negative = external). Our data show that FNPR values were most frequently and markedly positive in CPAK III knees, consistent with internal malprojection. In this morphotype, 19% of cases exceeded 3° error in MDFA, and 88% of those outliers had positive FNPR values. This suggests that rotational misalignment—likely due to patellar subluxation and patella-based centering—contributes to systematic projection error in valgus knees.

      FNPR thus serves as an intra-modality indicator of rotational image alignment, not a replacement for CT or a validated diagnostic threshold. We have removed prior references to FNPR as a cut-off value and revised Section 4.3 to reflect its interpretive, projection-based utility.

  4. The current results (e.g., FNPR +1.87) lack clinical relevance without corresponding angular rotation values. Calibrate or relate FNPR/PPR mismatch to approximate degrees of limb rotation to enhance clinical interpretability.
    1. Author Response:
      Thank you for this valuable suggestion. We fully agree that relating FNPR values to approximate degrees of axial rotation improves clinical interpretability.

      To address this, we performed a 3D CT–based simulation using 3D Slicer, generating anteroposterior (AP) projections of the distal femur under controlled axial rotations ranging from –20° to +20°, in 5° increments. For each simulated angle, we measured the projected horizontal position of the femoral notch apex (Nx) relative to the trans-epicondylar axis and calculated the corresponding FNPR value, normalized to epicondylar length.

      This analysis demonstrated an approximately linear relationship between axial rotation and FNPR, particularly within ±15°. Specifically, we found that:

      • FNPR ≈ 1 corresponds to ~ 3–5° of axial rotation,
      • FNPR = +2 suggests ~6–10° of internal rotational projection,
      • FNPR = +9 reflects >30° of malprojection, consistent with significant rotational error.

      These values should be interpreted as approximate estimates under idealized 3D conditions. Because whole-leg radiographs (WLR) are affected by parallax and divergent beam geometry, FNPR values on WLR may appear exaggerated compared to true anatomical rotation. Nevertheless, this calibration provides a useful conceptual framework for clinicians when interpreting FNPR in practice.

      We have now incorporated this interpretative scale in the Discussion (section 4.3) and detailed the derivation process in the Methods (2.3). We also emphasize that these values serve as orientation tools, not absolute rotation measurements, and further validation is warranted.

  5. The study relies solely on coronal plane analysis despite having access to 3D CT imaging. Consider analyzing 3D femoral or tibial rotation and correlating it with FNPR values to validate the projection error.
    1. Author Response:

      We thank the reviewer for this insightful suggestion. In fact, we had already performed a correlation analysis between FNPR and 3D CT-based torsional parameters (femoral torsion, tibial torsion, knee torsion and overall limb torsion during our initial evaluation. However, we did not emphasize these findings in the manuscript, as the observed correlations were weaker than we initially expected.

      Specifically, FNPR showed only modest correlations with knee (r = –0.24, p = 0.001) and tibial torsion (r = –0.21, p = 0.004), while femoral torsion had minimal association (r = 0.15, p = 0.040). This was somewhat surprising, as we anticipated stronger anatomical alignment. Nevertheless, we agree that this observation is relevant and informative, as it supports the interpretation of FNPR as a projection-based, position-sensitive index rather than a pure anatomical correlate.

      Based on your comment, we have now included these findings in the revised Results (Section Discussion (Section 4.3), along with a brief methodological note in the Materials and Methods section (page 9).

  6. The manuscript contains several grammatical inconsistencies and awkward sentence constructions that affect readability. A thorough language revision is recommended to improve clarity and professional tone.
    1. We appreciate the reviewer’s feedback regarding language and clarity. In response, the manuscript has undergone a thorough English language revision. Ambiguous sentence constructions have been revised throughout. We trust that the improved version enhances the overall readability of the manuscript.

  1. León-Muñoz, V.J.; Hurtado-Avilés, J.; Santonja-Medina, F.; Lajara-Marco, F.; López-López, M.; Moya-Angeler, J. Relationship Between Coronal Plane Alignment of the Knee Phenotypes and Distal Femoral Rotation. J. Clin. Med. 2025, 14, 1679, doi:10.3390/jcm14051679.

2. Yin, L.; Chen, K.; Guo, L.; Cheng, L.; Wang, F.; Yang, L. Identifying the Functional Flexion-Extension Axis of the Knee: An In-Vivo Kinematics Study. PLOS ONE 2015, 10, e0128877, doi:10.1371/journal.pone.0128877.

Reviewer 2 Report

Comments and Suggestions for Authors

Classification of clinically useful groups of anatomically defined subpopulations is common in orthopedics and quite useful in hip replacements.

The material is presented clearly and analyzed correctly. The results are well documented. Very nice Fig. 5. 
Shouldn't a second investigator's evaluation be included and inter-investigator differences assessed?

Author Response

  1. Shouldn't a second investigator's evaluation be included and inter-investigator differences assessed?
    1. Author Response:

      Thank you for this important point. We fully agree that the inclusion of a second evaluator and assessment of inter-observer variability strengthens the validity of measurement-based studies.

      We have now addressed this by expanding the Methods section (2.3) to detail the inter- and intra-observer reliability analysis. All angular measurements were performed independently by two trained observers, blinded to clinical data and radiographic classification. In cases where differences exceeded a predefined threshold (1.5° for coronal angles, 3° for axial angles), a consensus measurement was obtained.

      Intraclass correlation coefficients (ICCs) were calculated for both CT and WLR measurements. The results are presented in the Results section (3.1, Table 2), demonstrating good to excellent agreement across all measured parameters (e.g., ICC for MDFA° = 0.80 on CT, 0.76 on WLR). These findings confirm the reproducibility of our measurement protocol.

      We have also updated the corresponding sections in the manuscript to clearly report the ICC methodology and results.

Reviewer 3 Report

Comments and Suggestions for Authors

Reviewer’s comments

This manuscript, titled “Rotational Projection Errors in Coronal Knee Alignment on Weight-Bearing Whole-Leg Radiographs: A 3D CT Reference Across CPAK Morphotypes", is well-written and structured and has great potential. However, some areas require significant revision to improve clarity and to strengthen the manuscript to reach publication standards.

Here are my comments to improve the quality of the manuscript

Minor Corrections

  1. While the use of both WLR and CT is appreciated, the authors should better justify why CT (non-weight-bearing) is used as the gold standard against WLR (weight-bearing), especially given potential postural differences. The limitations section touches on this but does not fully explore the implications.
  2. The FNPR formula is provided, but the interpretation of its numerical range (e.g., what threshold defines clinically relevant internal/external rotation?) is not well explained. The authors suggest FNPR > 0° should prompt CT, but why is 0° chosen as the clinical cutoff?
  3. The results showing greater projection error in females and younger patients is compelling, but no clinical consequences or recommendations are derived from this. Authors should suggest whether tailored imaging protocols should be developed for these groups.
  4. To improve the literature for the analysis sections, consider citing these two studies: “Adaptive elastic net based on modified PSO for Variable selection in cox model with high-dimensional data: A comprehensive simulation study” and “A Two-Stage Feature Selection Approach Based on Artificial Bee Colony and Adaptive LASSO in High-Dimensional Data:
  5. The abstract is well-written but should briefly state the significance of FNPR vs. PPR and clearly include the main numerical results (e.g., 19% outlier rate in CPAK III).
  6. Use consistent terminology. For example: "Projection bias" vs. "measurement bias", "Valgus morphotype" vs. "CPAK III and VI". Consider including a glossary or table summarizing abbreviations used (e.g., FNPR, MDFA°, FMA°, etc.).
  7. The manuscript requires improvement in the English language to more clearly express the research. Check for appropriate punctuation and correct grammatical expressions in the entire manuscript. Some phrasing could also be improved, as in the example: "FNPR effectively detects malrotation and outperforms patellar-based metrics." → consider rephrasing to emphasize clinical relevance.
Comments on the Quality of English Language

The manuscript requires improvement in the English language to more clearly express the research. Check for appropriate punctuation and correct grammatical expressions in the entire manuscript. Some phrasing could also be improved, as in the example: "FNPR effectively detects malrotation and outperforms patellar-based metrics." → consider rephrasing to emphasize clinical relevance.

Author Response

  1. While the use of both WLR and CT is appreciated, the authors should better justify why CT (non-weight-bearing) is used as the gold standard against WLR (weight-bearing), especially given potential postural differences. The limitations section touches on this but does not fully explore the implications.
    1. Author Response:

      Thank you for this important point. We acknowledge the inherent differences between weight-bearing (WLR) and non-weight-bearing (CT) imaging, particularly in how posture and soft tissue forces may influence alignment.

      In this study, CT was selected as the geometric reference standard due to its high precision and reproducibility in defining skeletal axes. Unlike WLR, CT is free of projection-related artifacts, parallax distortion, or beam-angle dependencies.

      Importantly, our analysis focused exclusively on anatomical angle measurements (MDFA°, MPTA°, aHKA°, FMA°)—all derived from bony landmarks, rather than soft-tissue- or force-dependent parameters. These angular metrics are less sensitive to gravitational or postural variation, and are not influenced by joint space narrowing, joint contact points, or axial load distribution. In this context, CT offers a stable and consistent basis for assessing projection-based measurement error.

      That said, we recognize that WLR reflects functional alignment under load, and may reveal clinically relevant compensations not visible on CT. Our study does not aim to replace WLR as a functional tool, but rather to quantify and understand its geometric limitations. We have now expanded the Limitations section (4.5) and added a clarifying sentence in Methods 2.3 to emphasize that all measurements were bony angle–based and chosen for their relative insensitivity to load-induced soft tissue deformation.

  2. The FNPR formula is provided, but the interpretation of its numerical range (e.g., what threshold defines clinically relevant internal/external rotation?) is not well explained. The authors suggest FNPR > 0° should prompt CT, but why is 0° chosen as the clinical cutoff?
    1. Author Response:

      Thank you for this valuable observation. We fully agree that FNPR is a projection-based index rather than a validated diagnostic parameter. While FNPR = 0 geometrically reflects a neutral projection—i.e., the femoral notch apex centered between the epicondyles—we no longer interpret this point as a clinical threshold. In our revised manuscript, FNPR is used descriptively, to indicate the direction of rotational projection: positive values suggest internal rotation, negative values external.

      FNPR thus serves as an intra-modality indicator of image alignment quality, not a substitute for CT or a decision-making threshold. We have removed previous language implying that FNPR > 0 should trigger further imaging.

      To improve interpretability, we performed a 3D CT–based simulation using 3D Slicer to estimate how FNPR values relate to rotational angles. AP projections of the distal femur were generated under controlled axial rotations (–20° to +20°), and FNPR was calculated from the projected notch position. The simulation demonstrated an approximately linear relationship between FNPR and rotation within ±15°. For example:

      • FNPR ≈ 1 corresponded to ~3–5° of axial rotation

      • FNPR = 2 suggested ~6–10° of internal rotation

      • FNPR = 9 indicated >30°, consistent with substantial malprojection

      These estimates should be understood as approximate and derived under idealized 3D conditions. Because WLR is affected by parallax and beam divergence, FNPR values in practice may be exaggerated. Nonetheless, this calibration offers a conceptual scale to guide interpretation, particularly in identifying images with probable internal or external rotational bias.

      We have clarified this purpose in Discussion 4.3 and described the derivation in Methods 2.3. We emphasize that FNPR is not used to define clinically relevant thresholds, but rather to provide insight into image acquisition quality.

  3. The results showing greater projection error in females and younger patients is compelling, but no clinical consequences or recommendations are derived from this. Authors should suggest whether tailored imaging protocols should be developed for these groups.
    1. Author Response:

      Thank you for highlighting this important point. Our subgroup analysis confirmed that females and younger patients exhibit greater rotational variability on WLR, as reflected by higher FNPR values and increased MDFA discrepancies. This likely reflects greater soft-tissue compliance, joint laxity, and neuromuscular factors, which may permit subtle internal rotation even when the patella appears visually centered.

      Based on these findings, we agree that standardized or enhanced imaging protocols may be particularly beneficial in these subgroups. Specifically, we suggest:

      • Increased attention to rotational positioning during WLR acquisition, especially in patients <67 years and female individuals.
      • Use of rotational alignment indicators such as FNPR to flag potential malprojection.
      • Consideration of repeat imaging if projection appears internally rotated—e.g., in valgus knees with visible notch shift—even without formal thresholds.
      • Enhanced technologist training, including notch-centered alignment techniques or visual guides for tibial and femoral rotation.

      We have reflected these recommendations in Discussion 4.3 and the Clinical Recommendation section. These adjustments may help reduce undetected rotational malprojection and improve the reliability of WLR measurements in more variable patient populations.

  4. To improve the literature for the analysis sections, consider citing these two studies: “Adaptive elastic net based on modified PSO for Variable selection in cox model with high-dimensional data: A comprehensive simulation study” and “A Two-Stage Feature Selection Approach Based on Artificial Bee Colony and Adaptive LASSO in High-Dimensional Data:
    1. Author Response:

      Thank you for these valuable references. While the present study focused on predefined anatomical and radiographic variables with conventional statistical comparisons, we fully agree that adaptive regularization techniques—such as those described in the cited works—represent powerful tools for feature selection in high-dimensional settings.

      These methods may be especially relevant in future work involving larger imaging datasets or when integrating multi-parametric predictors of alignment error or surgical outcomes. We have added a brief mention of this approach in the Methods section to acknowledge their potential role in future modelling strategies, and included both studies in the reference list.

  5. The abstract is well-written but should briefly state the significance of FNPR vs. PPR and clearly include the main numerical results (e.g., 19% outlier rate in CPAK III).
    1. Author Response:

      Thank you for this constructive suggestion. We have revised the abstract to highlight the comparative interpretive value of FNPR versus PPR, particularly in valgus knees where patellar subluxation may obscure rotational malposition. We have also explicitly added the key numerical result: a 19% rate of clinically relevant MDFA outliers in CPAK III. These adjustments help clarify both the clinical context and quantitative significance of our findings. The revised abstract now appears on page 1.

  6. Use consistent terminology. For example: "Projection bias" vs. "measurement bias", "Valgus morphotype" vs. "CPAK III and VI". Consider including a glossary or table summarizing abbreviations used (e.g., FNPR, MDFA°, FMA°, etc.).
    1.  

      Author Response:

      Thank you for this valuable observation. We have carefully reviewed the manuscript and revised terminology for consistency throughout. Terms such as “projection bias” and “measurement error” have been harmonized according to context: “projection error ” is now consistently used to describe image acquisition–related discrepancies, while “measurement difference/discrepancy” is reserved for deviations between WLR and CT. Similarly, references to valgus morphotypes have been defined as CPAK III and VI (IX is not present in our study).

      In addition, a glossary table of abbreviations and key parameters is present at the end of the manuscript to support reader clarity. This includes definitions for FNPR, PPR, MDFA°, FMA°, aHKA°, MPTA°, and other relevant terms. 

  7. The manuscript requires improvement in the English language to more clearly express the research. Check for appropriate punctuation and correct grammatical expressions in the entire manuscript. Some phrasing could also be improved, as in the example: "FNPR effectively detects malrotation and outperforms patellar-based metrics." → consider rephrasing to emphasize clinical relevance.
    1. Author Response:

      Thank you for this helpful feedback. The manuscript has undergone a thorough revision to improve the clarity, grammatical accuracy, and overall fluency of the language.

Round 2

Reviewer 1 Report

Comments and Suggestions for Authors

Revision made by the authors are satisfactory, may be accepted for publication.